# An Updated Perspective on the Aromatic Metabolic Pathways of Plant-Derived Homocyclic Aromatic Compounds in *Aspergillus niger*

**DOI:** 10.3390/microorganisms13081718

**Published:** 2025-07-22

**Authors:** Ronnie J. M. Lubbers

**Affiliations:** Fungal Genetics and Biotechnology Group, Institute of Biology, Leiden University, Sylviusweg 72, 2333 BE Leiden, The Netherlands; r.j.m.lubbers@biology.leidenuniv.nl

**Keywords:** aromatic metabolism, aspergilli, benzoic acids, cinnamic acids, phenolic compounds

## Abstract

Aromatic compounds are vital in both natural and synthetic chemistry, and they are traditionally sourced from non-renewable petrochemicals. However, plant biomass, particularly lignin, offers a renewable alternative source of aromatic compounds. Lignin, a complex polymer found in plant cell walls, is the largest renewable source of aromatic compounds, though its degradation remains challenging. Lignin can be chemically degraded through oxidation, acid hydrolysis or solvolysis. As an alternative, microorganisms, including fungi, could offer a sustainable alternative for breaking down lignin. The aromatic compounds released from lignin, by either microbial, chemical or enzymatic degradation, can be used by microorganisms to produce valuable compounds. Fungi possess unique enzymes capable of converting aromatic compounds derived from lignin or other sources into chemical building blocks that can be used in several industries. However, their aromatic metabolic pathways are poorly studied compared to bacterial systems. In the past, only a handful of genes and enzymes involved in the aromatic metabolic pathways had been identified. Recent advances in genomics, proteomics, and metabolic engineering are helping to reveal these metabolic pathways and identify the involved genes. This review highlights recent progress in understanding fungal aromatic metabolism, focusing on how *Aspergillus niger* converts plant-derived aromatic compounds into potentially useful products and the versatility of aromatic metabolism within the *Aspergillus* genus. Addressing the current knowledge gaps in terms of fungal pathways could unlock their potential for use in sustainable technologies, promoting eco-friendly production of chemical building blocks from renewable resources or bioremediation.

## 1. Introduction

Aromatic compounds play crucial roles in plants and provide structural support (e.g., lignin), furnish a defense against herbivores and pathogens (e.g., phenolics), and contribute to pigmentation, fragrance, and flavor, aiding in pollination and protection against environmental stress. Additionally, they are also involved in metabolic processes and allelopathy, helping plants compete and thrive in their ecosystems [1]. Aromatic compounds are also prevalent in everyday life and are found in a wide range of man-made products, such as beverages, cosmetics, foods, fuels, lubricants, medicines, paints, and plastics [2]. Most commercial aromatic compounds utilized today are synthesized from base chemicals like benzene, toluene and xylene, which are typically extracted from nonrenewable fossil resources. Therefore, developing methods to utilize the wealth of natural aromatic compounds and producing those from renewable sources is essential for sustainable practices and reducing dependence on fossil fuels.

One of the most promising renewable sources of aromatic compounds is the complex plant polymer lignin. Between 15 and 30% of the dry weight of plant biomass consists of lignin, depending on the species. Lignin is mostly built from the aromatic compounds coniferyl alcohol (guaiacyl (G)-units), *para*-coumaryl alcohol (*p*-hydroxyphenyl (H)-units) and sinapyl alcohol (syringyl (S)-units), derived from hydroxycinnamic acids, and the ratios of building blocks used in lignin vary per plant species. Additionally, different building blocks, such as caffeoyl alcohol and vanillyl alcohol, have been observed in lignin [3]. Currently, much research is performed on the utilization of lignin through chemical, enzymatic and biotransformation approaches [4,5,6,7]. Degradation of lignin results in complex mixtures of monomeric, dimeric and polymeric aromatic compounds. In this review, lignin degradation or depolymerization through chemical, enzymatic or microbial means will not be discussed since this topic has been reviewed extensively over the past decade.

In addition to lignin, plants produce a wide array of aromatic compounds through the shikimate and phenylpropanoid pathways, which are central to secondary metabolism. These pathways lead to the biosynthesis of flavonoids, tannins, phenolic acids, and various alkaloids. Aromatic compounds can also be found in plants in other complexes, such as the hydroxycinnamic acids, ferulic acid (3-methoxy-4-hydroxycinnamic acid) and *p*-coumaric acid (4-hydroxycinnamic acid), which can be linked to complex plant cell wall structures like hemi-cellulose and pectin and have ester linkages with arabinose and galactose [8,9,10]. Many monomeric aromatic compounds can also be found freely in plants; for example, vanillin in vanilla pods, cinnamaldehyde in cinnamon, benzaldehyde in almonds and caffeic acid in coffee beans. Due to the wide range of aromatic compounds found in plants, mainly monomeric aromatic compounds derived from lignin will be discussed.

Fungi are able to degrade and metabolize a wide range of monomeric aromatic compounds efficiently, but the aromatic metabolic pathways in fungi are poorly studied. Moreover, most existing studies focus on different fungal species, making it challenging to determine which specific pathways are present within a single species. Hence, a complete fungal aromatic metabolic pathway overview is not available. To address these gaps, a comparison of the aromatic metabolic pathways in bacteria, yeast, and fungi was conducted previously [11], revealing that many of these pathways are shared among them but unique pathways also exist. Before 2020, the best-described aromatic metabolic pathways were from *Aspergillus japonicus* [12]. However, this study was before the genome era, so no genes were identified and it contains many gaps. Despite the fact that many fungal genomes are now sequenced and published, the number of identified fungal genes encoding enzymes converting aromatic compounds is remarkably low [11,13]. It was also revealed that unique and unexplored aromatic metabolic pathways exist in fungi [11].

*Aspergillus* species are filamentous, cosmopolitan, and ubiquitous fungi commonly found in soil, plant debris, and indoor air environments. These fungi play crucial roles in the decomposition of organic matter, helping recycle carbon and nutrients in ecosystems. Currently, over 185 *Aspergillus* species have been identified, and this number continues to grow. In addition, these fungal species have been used in many applications; for example, *Aspergillus niger*, which is used for citric acid and enzyme production, such as amylases, and *Aspergillus oryzae* and *Aspergillus sojae*, which are widely used in fermentation processes for making sake and soy sauce. On the other hand, certain *Aspergillus* species are pathogenic, especially to immunocompromised individuals. The fungus *Aspergillus fumigatus* is known to be the primary cause of aspergillosis. *Aspergillus* species are known to have large arsenals of enzymes that can degrade complex biomass, polymers and molecules. Despite not being known for their lignin degradation capabilities, lignin-degrading *Aspergillus* species have been recently discovered and isolated [14,15,16]. Recently, multiple genes and enzymes involved in the metabolism of monomeric aromatic compounds, mainly in *Aspergillus* species from the *Nigri* section [17], have been identified (Table 1). The identification of genes associated with aromatic metabolic pathways allows for improved predictions and correlations regarding which pathways are present in fungi. In this review, the recent advances and insights in terms of the monomeric aromatic metabolic pathways in fungi from the past few years, with a main focus on *A. niger*, are highlighted and discussed.

## 2. Metabolism of Hydroxycinnamic Acids

The monomeric aromatic compounds ferulic acid (3-methoxy-4-hydroxycinnamic acid) and *p*-coumaric acid (*p*-hydroxycinnamic acid) are hydroxycinnamic acids that are part of the complex sugars and are linked to arabinose and galactose residues via ester linkages [8,9,10]. These linkages are part of the plant cell wall and contribute to the cell wall rigidity and aid in the plants’ resistance against environmental stress and pathogens. Multiple fungi are able to hydrolyze the ester bonds between hydroxycinnamic acids and polysaccharides using feruloyl esterases (Faes). These enzymes play a crucial role in the degradation to complex plant biomass, and it has been shown that Faes can be used to release ferulic acid from agricultural waste streams such as wheat bran, maize bran and sugar beet pulp [37,38,39]. Faes have been identified and studied in many fungi but mainly in the Aspergilli: *A. niger*, *A. oryzae* and *Aspergillus nidulans* [40]. At this moment, three Faes have been identified and characterized in *A. niger* [41,42,43]. Homologs of feruloyl esterases have been observed and studied in multiple fungi, including *Aspergillus*, *Fusarium*, *Penicillium*, *Pleurotus*, *Trichoderma*, and *Talaromyces* species [40].

Multiple pathways that can convert ferulic acid have been described. *A. oryzae* can release ferulic acid from the cell wall of rice endosperm during brewing processes. After the release, ferulic acid is decarboxylated to 4-vinylguaiacol (Figure 1) [44]. 4-vinylguaiacol can lead to an undesired flavor in distilled products such as beer, awamori and sake [31,32,33,45,46]. In *Aspergillus luchuensis*, decarboxylation of hydroxycinnamic acids to the vinyl form is catalyzed by phenolic acid decarboxylase (Pad). This conversion is also observed in the *A. niger* diploid strain DAR2 (derived from *A. niger* C28B25). This strain is able to decarboxylate ferulic acid to 4-vinylguaiacol and further to vanillic acid, possibly with vanillin as an intermediate (Figure 1) [47]. A close homolog of AlPad is present in *A. niger* (NRRL3_08440) and could also be involved in the decarboxylation of ferulic acid. This pathway has also been observed in other fungi, such as *Fusarium solani* (Mart.) Sacc., *Colletotrichum gloeosporioides*, *Isaria farinosa* and many yeast species [45,46,48,49], indicating that this pathway is relatively common. Interestingly, *A. niger* C28B25, the parental strain of DAR2, degrades ferulic acid to vanillic acid without the non-oxidative decarboxylation step to vinyl-guaiacol, indicating that other pathways are present in *A. niger* [47].

Decarboxylation of hydroxycinnamic acid into its vinyl form has been observed in many fungi. In *Saccharomyces cerevisiae*, hydroxycinnamic acids and cinnamic acid are decarboxylated by ferulic acid decarboxylase 1 (Fdc1) and flavin prenyltransferase 1 (Pad1), respectively [25,50]. Homologs of Fdc1 and Pad1 are found in multiple *Aspergillus* species and have been shown to perform this catalysis in vitro [26,51]. However, deletion of these genes in *A. niger* resulted in the cessation of growth on cinnamic acid, but no phenotypes were observed on the hydroxycinnamic acids: ferulic acid, *p*-coumaric acid and caffeic acid [26]. This provided new indications that these enzymes are involved in vivo in cinnamic acid decarboxylation but not hydroxycinnamic acid decarboxylation [26]. In addition, this was also supported by transcriptome data showing that *fdcA* and *padA* are not induced when grown on caffeic acid, ferulic acid or *p*-coumaric acid. Therefore, the FdcA of *A. niger* was renamed cinnamic acid decarboxylase A (CdcA) [26].

Recently, it has been shown that *A. niger* degrades the hydroxycinnamic acids, caffeic acid, ferulic acid, *p*-coumaric acid and *meta*-coumaric acid (3-hydroxycinnamic acid), through the peroxisomal CoA-dependent beta-oxidative metabolic pathway, resulting in the formation of protocatechuic acid (3,4-dihydroxybenzoic acid), vanillic acid (4-hydroxy-3-methoxybenzoic acid), *p*-hydroxybenzoic acid and *m*-hydroxybenzoic acid, respectively (Figure 1) [36]. Also, dihydrocaffeic acid (3-(3,4-dihydroxyphenyl)propanoic acid) and phloretic acid (3-(4-hydroxyphenyl)propanoic acid) are degraded through this pathway [36]. The CoA-dependent beta-oxidative metabolic pathway is well known for the degradation of fatty acids but was not previously known to be involved in the degradation of hydroxycinnamic acids. Interestingly, also in plants, it has been shown that hydroxycinnamic acid can be degraded through this pathway [52]. Multiple genes involved in the degradation of hydroxycinnamic acids in *A. niger* were identified using whole-genome transcriptome data [36].

In *A. niger*, the first step in the CoA-dependent beta-oxidative metabolic pathway is catalyzed by the hydroxycinnamate-CoA synthase (HscA), and the deletion of it results in strongly reduced growth on hydroxycinnamic acids but appears not to be required for fatty acid degradation [36]. The second and third steps in this pathway are catalyzed by fatty acid oxidase A (FoxA) and observed to be involved in the fatty acid degradation in multiple fungal species [36,53,54,55,56,57,58]. The fourth step is catalyzed by 3-ketoacyl-CoA thiolase (KatA) (Figure 1). Deletion of these related genes resulted in clear phenotypes on several hydroxycinnamic acids; however, growth was not stopped on ferulic acid or *p*-coumaric acid, indicating that another pathway, functional redundancy, or alternative gene is present in *A. niger* [36]. The final step in the CoA-dependent oxidative pathway is proposed to be catalyzed by a thioesterase. Four thioesterases (TheA, TheB, TheC and TheD) have been identified and the deletion of these individual genes did not result in reduced growth on ferulic acid, *p*-coumaric acid and caffeic acid. Deletion of *theB* results in a growth reduction on *m*-coumaric acid, but the growth recovers over time and it was suggested that there is functional redundancy in the final step. The redundancy of the thioesterases could be further studied by making a quadruple deletion of the genes and analyzing the growth on hydroxycinnamic acids. Interestingly, deletion of the CoA-dependent beta-oxidative genes did not result in reduced growth on sinapic acid (3,5-dimethoxy-4-hydroxycinnamic acid) or cinnamic acid, indicating that they are degraded through another pathway(s) in *A. niger*.

In *A. niger*, *o*-coumaric acid (2-hydroxycinnamic acid) can be converted into 4-hydroxycoumarin but appears not to be further degraded [59]. In *A. fumigatus*, melilotic acid (3-(2-hydroxyphenyl)propanoic acid) is converted through esterification to *o*-coumaric acid and further to 4-hydroxycoumarin and dicoumarol [60,61]. Interestingly, a CoA-dependent beta-oxidative pathway was suggested for the conversion of *o*-coumaric acid to 4-hydroxycoumarin with *o*-coumaryl-SCoA, *o*-hydroxyphenyl-β-hydroxypropionyl-SCoA, and *o*-hydroxyphenyl-β-ketopropionic acid-SCoA as intermediates [61]. As mentioned before, in *A. niger*, dihydrocaffeic acid (3-(3,4-dihydroxyphenyl)propanoic acid) and phloretic acid (3-(4-hydroxyphenyl)propanoic acid) are converted through the CoA-dependent beta-oxidative pathway, and it is possible that the compounds are esterified to caffeic acid and *p*-coumaric acid, respectively. However, the genes involved in this pathway still need to be identified.

An alternative metabolic pathway for the metabolism of hydroxycinnamic acid is a CoA-independent non-oxidative pathway that has been observed in *A. flavus* [62]. *p*-coumaric acid is converted to β-hydroxy-(*p*-hydroxyphenyl)propionic acid and *p*-hydroxybenzoylacetic acid, followed by conversion to *p*-hydroxybenzoic acid. In *A. luchuensis*, it is proposed that ferulic acid is converted to vanillin through a CoA-dependent non-oxidative pathway with feruloyl-CoA and 4-hydroxy-3-methoxyphenyl-β-hydroxypropionyl-CoA (HMPKP-SCoA) as intermediates [33]. In *A. brasiliensis* ATCC16404, the 2′ position of the aliphatic chain of ferulic acid is methoxylated to (E)-3-(4-hydroxy-3-methoxyphenyl)-2-methoxyacrylic acid [63].

## 3. Metabolism of Cinnamic Acid

The decarboxylation of cinnamic acid has been observed in multiple *Aspergilli* and *Trichoderma* species, including *A. niger*, *A. japonicus*, *A. flavus*, *A. oryzae*, *A. wentii*, *T. reesei*, *T. viride*, and *T. koningii*, and also in *Penicillium* species and the yeast *S. cerevisiae* [12,25,26,64,65,66]. As mentioned before, *A. niger* decarboxylates cinnamic acid to styrene via CdcA and PadA (Figure 2). Both genes are also involved in the degradation of sorbic acid and are regulated by the sorbic acid regulator A (SdrA) [25,26,27]. Interestingly, all three genes are clustered on the genome of many Aspergilli and a strong correlation was found in Aspergilli species between the ability to grow on cinnamic acid and sorbic acid and this gene cluster [26]. For example, *A. nidulans* is unable to grow on cinnamic acid due to a mutation resulting in truncated *padA*.

In *A. japonicus* and *A. niger*, cinnamic acid can also be converted to benzoic acid [12,26,28]. Deletion of benzoate-4-monooxygenase (*bphA*), *p*-hydroxy-*m*-hydroxylase (*phhA*)*,* protocatechuate 3,4-dioxygenase (*prcA*) or the combination of protocatechuate hydroxylase (*phyA*)/*prcA* or hydroxyquinol 1,2-dioxygenase (*hqdA*)/*prcA* in *A. niger* results in reduced growth on cinnamic acid, indicating that it is converted to benzoic acid [28,29]. In addition, *A. niger* Δ*prcA*Δ*phyA* transformant grown on cinnamic acid and cinnamyl alcohol resulted in the accumulation of protocatechuic acid [28]. Currently, it remains unknown whether styrene is an intermediate in this conversion pathway or an alternative cinnamic acid metabolic pathway is present in *A. niger*. However, in other fungi, cinnamic acid pathways toward benzoic acid and *p*-hydroxybenzoic acid have been observed. In *Phomopsis liquidambari*, it is suggested that cinnamic acid is decarboxylated to styrene, followed by conversion to benzaldehyde and benzoic acid by a putative laccase [67]. In the yeast *Yarrowia lipolytica* OKYL029, cinnamic acid is hydroxylated to *p*-coumaric acid and further converted to *p*-hydroxybenzoic acid [68]. Deletion of a cytochrome P450 sharing homology with trans-cinnamate 4-monooxygenases (YALI1_B28430g, TCM1) resulted in a blockage of *p*-coumaric acid formation; however, cinnamic acid was still converted through an unknown pathway. The conversion of cinnamic acid to *p*-coumaric acid was also suggested in *A. japonicus* [12] and observed in *A. niger* [59]. However, in the latter, melilotic acid (3-(2-hydroxyphenyl)propanoic acid), *o*-coumaric acid and *p*-hydroxybenzoic acid were also detected (Figure 2) [59].

In several fungi, other alternative cinnamic acid metabolic pathways have been observed. In *Neurospora crassa* and *Mucor* sp. JX23, cinnamic acid is converted to acetophenone [69,70]. In *A. japonicus* and *Schizophyllum commune*, cinnamic acid can be reduced to cinnamaldehyde and cinnamyl alcohol [12,71]. The genes encoding these enzymes still remain to be identified.

## 4. Metabolism of Benzoic Acid and Related Aromatic Compounds

In fungi, multiple aromatic compounds are metabolized and funneled toward protocatechuic acid [11,28]. In *A. niger*, all the *p*-hydroxyphenyl and 3,4-dihydroxyphenyl units, which are derived from the H-unit monolignol, are converted to protocatechuic acid, indicating that this pathway plays an important role in the degradation of hydroxylated aromatic compounds in fungi.

One of the aromatic compounds that is converted to protocatechuic acid is benzoic acid (benzene carboxylic acid), a commonly used food preservative with anti-microbial properties [72,73]. In multiple *Aspergillus* species, benzoic acid can be hydroxylated by benzoate-4-monooxygenase (BphA) to *p*-hydroxybenzoic acid [18,19,20,29,74,75]. This pathway is continued by the conversion of *p*-hydroxybenzoic acid into protocatechuic acid catalyzed by PhhA, with further conversion into 3-carboxy-*cis*,*cis*-muconic acid by PrcA [21,29]. Interestingly, deletion of *prcA* in *A. niger* did not halt growth on protocatechuic acid, indicating that an alternative pathway is present. Protocatechuic acid can also be converted to hydroxyquinol (1,2,4-trihydroxybenzene) by protocatechuate hydroxylase (PhyA) and is further converted by hydroxyquinol 1,2-dioxygenase (HqdA) [28,29]. Deletion of *phyA* did not result in a growth reduction on protocatechuic acid, revealing that this metabolic pathway may have a minor role in protocatechuic acid metabolism, as also observed in *A. nidulans* [76]. In *A. niger*, benzoic acid can also be hydroxylated to *m*-hydroxybenzoic acid and further to protocatechuic acid [77]. BphA is not able to *meta*-hydroxylate benzoic acid to *m*-hydroxybenzoic acid [19], while deletion of *bphA* results in severely reduced growth on benzoic acid [29]. Therefore, the conversion of benzoic acid into *m*-hydroxybenzoic acid is likely to be a minor pathway in *A. niger*.

With this knowledge, the first reported *A. niger* protocatechuic-acid-accumulating cell factory was constructed [28]. Protocatechuic acid can be used as a precursor for *cis,cis*-muconic acid for nylon, polyurethane, and PET or vanillin for the food industry. This cell factory is able to accumulate protocatechuic acid from many aromatic compounds, such as benzoic acid, benzaldehyde, benzyl alcohol, *p*-anisic acid, *p*-anisaldehyde, *p*-anisyl alcohol, *p*-hydroxybenzoic acid, *p*-hydroxybenzaldehyde, *p*-hydroxybenzyl alcohol, *m*-hydroxybenzoic acid, protocatechuic aldehyde, *p*-coumaric acid, caffeic acid, cinnamic acid and cinnamyl alcohol. High protocatechuic acid accumulation rates, up to a 90–99% molar yield, were obtained from most of the tested aromatic compounds. It has to be noted that the experiments were performed at a lab scale. Interestingly, accumulation of protocatechuic acid was not observed on ferulic acid, vanillin, veratric acid, *p*-cresol, or anethole, indicating that these are converted through alternative pathways. This study contributed greatly to mapping out which aromatic compounds are ultimately converted into protocatechuic acid in *A. niger* (Figure 3).

Several aromatic compounds were not tested for protocatechuic acid accumulation with the *A. niger* cell factory [28] but were observed in other studies to be converted into protocatechuic acid. In *A. niger*, the chlorinated derivatives 2-chlorobenzoate and 3-chlorobenzoate are both observed to be converted into *p*-hydroxybenzoic acid and further into protocatechuic acid [78]. Mandelic acid is also converted into protocatechuic acid, with benzoyl formate, benzaldehyde, benzoic acid and *p*-hydroxybenzoic acid as intermediates [79]. The polyethylene terephthalate (PET)-derived compounds dimethyl-terephthalate, monomethyl-terephthalate and terephthalate are all suggested to be converted into protocatechuic acid [80]. The protocatechuic-acid-accumulating strain could be used to funnel complex mixtures of aromatic compounds into protocatechuic acid. In *A. flavus*, conversion of *p*-coumaryl alcohol into *p*-coumaric aldehyde, followed by conversion into *p*-coumaric acid, followed by conversion into *p*-hydroxybenzoic acid and protocatechuic acid, was observed [62]. Interestingly, *A. japonicus* can convert veratric acid into protocatechuic acid [12], while in *A. niger*, veratric acid is converted into vanillic acid [81]. *A. fumigatus* is able to degrade *p*-cresol, which is suggested to subsequently be converted into *p*-hydroxybenzyl alcohol, followed by *p*-hydroxybenzaldehyde and *p*-hydroxybenzoic acid, and finally, into protocatechuic acid [34]. In another suggested pathway, *p*-cresol is converted into 4-methylcatechol, (3,4-dihydroxybenzyl alcohol), protocatechualdehyde and protocatechuic acid, which is then converted by *prcA* into 3-oxoadipate. No evidence was found in *A. niger* that *p*-cresol is converted through these pathways [28]. In *A. japonicus*, *p*-anisic acid can also be reduced to the corresponding aldehyde and alcohol form [12], but this was not observed in *A. niger* [28]. In *A. sojae*, protocatechuic acid is decarboxylated to catechol [75].

Despite the fact that most *meta*- and *para*-hydroxylated benzoic acids are converted toward protocatechuic acid, several exceptions have been observed (Figure 3). *A. flavus* and *A. niger* are also able to reduce benzoic acid to benzyl alcohol, possibly with benzaldehyde as an intermediate [82,83].

## 5. Metabolism of Guaiacyl Units and Related Aromatic Compounds

Guaiacyl units, a key building block of lignin in gymnosperms (softwoods), feature a methoxy group at the *meta*-position and a hydroxyl group at the *para*-position on their aromatic ring (Figure 4). Coniferyl alcohol, ferulic acid, guaiacol, vanillic acid and vanillin are all guaiacyl units. As mentioned previously, in *A. niger*, ferulic acid is converted into vanillic acid through the CoA-dependent beta-oxidative metabolic pathway. Recently, the gene and enzyme vanillate hydroxylase A (VhyA), involved in the conversion of vanillic acid into methoxyhydroquinone, was identified [81]. This pathway was also observed in *A. flavus*, *Paecilomyces variotii* and *Sporotrichum pulverulentum* [82,84,85]. Methoxyhydroquinone is further processed through ring cleavage by methoxyhydroquinone 1,2-dioxygenase (MhdA) [81]. Deletion of *vhyA* in *A. niger* showed that, when grown on coniferyl alcohol, ferulic acid, vanillin, vanillyl alcohol, veratryl alcohol, veratric aldehyde or veratric acid, the fungus converts these substrates into vanillic acid, which subsequently accumulates due to the deletion of *vhyA* [81].

In *A. niger*, vanillin is converted into vanillic acid by vanillin dehydrogenase (VdhA) (Figure 4). Deletion of *vdhA* halted growth on vanillin and vanillin did not accumulate in the Δ*vdhA* mutant when grown on ferulic acid or coniferyl alcohol, indicating that ferulic acid is not degraded to vanillin in *A. niger* [35,81]. It has been shown in *A. niger* and *A. japonicus* that vanillin can also be reduced to vanillyl alcohol [12,81]. In *A. luchuensis*, vanillin is converted into vanillic acid but is also converted into vanillin-glucoside [33]. *Aspergillus carbonarius* and several *Trichoderma* species have been observed to produce vanillin from vanillic acid, while none of the other tested *Aspergillus* and *Penicillium* species were able to perform this conversion, which indicates that they are lacking a vanillate reductase [86]. Demethylation of vanillin to *p*-hydroxybenzaldehyde was observed in *A. japonicus* [12].

Other pathways have been observed in fungi in which vanillic acid is decarboxylated to guaiacol, protocatechuic acid, vanillin and/or vanillyl alcohol [86]. Interestingly, *A. flavus*, *A. niger* and *A. nidulans* are only able to convert vanillic acid into methoxyhydroquinone, while multiple *Aspergillus* and *Penicillium* species are able to convert vanillic acid into methoxyhydroquinone but also into protocatechuic acid or guaiacol.

In *A. japonicus*, it was suggested that veratric acid is demethylated to vanillic acid, followed by an additional demethylation to protocatechuic acid (Figure 4) [12]. The conversion of vanillic acid into protocatechuic acid has also been reported in *F. solani* [48]. In *A. niger*, veratric acid, veratric aldehyde and veratryl alcohol are converted into vanillic acid, but the conversion toward protocatechuic acid was not observed [28,81]. The gene encoding veratric acid demethylase remains to be identified. As observed with other aromatic acids, reduction to its aldehyde and alcohol form has been observed, where both *A. flavus* and *A. japonicus* are able to reduce veratric acid toward veratric aldehyde and veratryl alcohol [12,82].

## 6. Conversion of Syringyl Units

While syringyl units, in combination with guaiacyl units, are the predominant building blocks of lignin in hardwoods, the degradation of syringyl units by fungi remains less understood. Syringyl units, such as sinapyl alcohol, sinapic acid, syringic acid and syringic aldehyde, feature two methoxy groups at both *meta*-positions and a hydroxyl group at the *para*-position on their aromatic ring (Figure 5). It is observed that *A. niger* can degrade sinapic acid (3,5-dimethoxy-4-hydroxycinnamic acid) and syringic acid (3,5-dimethoxy-4-hydroxybenzoic acid) (Lubbers et al., unpublished data). However, it remains unknown how these are degraded. Sinapic acid is not degraded through the CoA-dependent beta-oxidative pathway since deletion of *hcsA*, *foxA*, or *katA* did not result in any growth phenotype [36]. No other new studies have become available concerning the degradation of syringyl units by Aspergilli and therefore the metabolic pathways remain unknown.

## 7. Gallic Acid Metabolic Pathways

Gallic acid (3,4,5-trihydroxybenzoic acid) is a commonly observed metabolite in the degradation of syringyl units [11] and is part of the polymeric aromatic compound tannic acid. It has been shown that many *Aspergillus* species can release gallic acid from tannic acid using tannases [87,88,89]. Gallic acid has anti-oxidative and anti-microbial properties and is applied in the cosmetic, food and beverage, agricultural and pharmaceutical industries [90].

Currently, not many gallic acid metabolic pathways have been observed in fungi [11]. In *A. oryzae*, gallic acid is converted into pyrogallol (1,2,3-trihydroxybenzene), progallin A (ethyl 3,4,5-trihydroxybenzoate) and methyl gallate (Figure 5) [91]. It was suggested that these compounds were further processed by a ring-opening reaction. In *A. niger* and *A. nidulans*, gallic acid is converted into 1,2,3,5-tetrahydroxybenzene, catalyzed by PhyA [76,92]. Deletion of 17 putative dioxygenase genes, including *prcA*, *hqdA*, and *crcA*, did not impair growth on gallic acid, suggesting that gallic acid is metabolized through an alternative pathway [92]. In *A. nidulans*, it has been proposed that 1,2,3,5-tetrahydroxybenzene is further converted into 5-hydroxydienelactone by a DUF3500-containing protein (AN10530) functioning as a putative dioxygenase [76]. Homologs of this gene are present in *A. niger* and remain to be studied.

## 8. Salicylic Acid Metabolic Pathways

Salicylic acid (2-hydroxybenzoic acid) is a well-known signaling molecule in plants and is involved in the defense against pathogens [93]. Currently, only a limited number of fungal salicylic acid metabolic pathways have been characterized. The most commonly observed conversion is the hydroxylation of the carboxylic group of salicylic acid, resulting in the formation of catechol (1,2-dihydroxybenzene). This conversion is catalyzed by the salicylic acid hydroxylase (ShyA) [22]. This conversion was observed in several Aspergilli, including *A. niger*, *A. japonicus*, *A. nidulans* and *A. terreus*, but also in other fungi, such as *Epichloë festucae* and *Sclerotinia sclerotiorum* (Figure 6) [12,22,76,94,95,96]. Alternatively, it has been proposed in *A. nidulans* and *Fusarium graminearum* that salicylic acid can also be hydroxylated to 2,3-dihydroxybenzoic acid (*o*-pyrocatechuic acid) [76,97], followed by a decarboxylation step to catechol catalyzed by 2,3-dihydroxybenzoic acid decarboxylase (DhbA) [22,23,24]. In *A. niger*, no evidence was found of the hydroxylation of salicylic acid to 2,3-dihydroxybenzoic acid since the deletion of *shyA* resulted in the cessation of growth on salicylic acid and deletion of *dhbA* did not result in reduced growth on salicylic acid [22]. Deletion of catechol 1,2-dioxygenase (CrcA) revealed that salicylic acid, 2,3-dihydroxybenzoic acid and catechol are converted toward *cis,cis*-muconic acid. *Cis-cis*-muconic acid is an interesting compound since it a valuable chemical building block that can be used for polymer and drug production [98]. In *A. fumigatus*, phenol can be *ortho*-hydroxylated to catechol and *cis,cis*-muconic acid but also *para*-hydroxylated to hydroquinone (1,4-dihydoxybenzene) and hydroxyquinol [34].

Recently, an additional salicylic metabolic pathway was observed in *A. terreus* in which salicylic acid is hydroxylated to gentisic acid (2,5-dihydroxybenzoic acid) (Figure 6) [94]. Gentisic acid is further converted into maleylpyruvate by a putative gentisate 1,2-dioxygenase (ATEG_06714). In *N. crassa*, salicylic acid can also be reduced to salicylic aldehyde and salicyl alcohol [99]. These pathways have not been observed in Aspergilli.

## 9. Ring Cleavage Pathways

The cleavage of the aromatic ring is a critical step in the detoxification and utilization of aromatic compounds as a carbon source. In fungi, most aromatic compounds are mainly converted toward catechol, hydroxyquinol, gallic acid, gentisic acid, protocatechuic acid or pyrogallol. These aromatic compounds are targeted by dioxygenases that catalyze the ring cleavage. The cleaved compounds are converted in multiple steps into acetyl-CoA, fumarate, oxaloacetate, pyruvate, or succinate and enter the TCA cycle [11,100]. In fungi, most aromatic compounds are cleaved by intradiol dioxygenases and use non-heme Fe(III) to cleave the aromatic nucleus *ortho* to the hydroxyl substituents [101]. Extradiol dioxygenases using non-heme Fe(II) or other divalent metal ions to cleave the aromatic nucleus *meta* to the hydroxyl substituents are less common in fungi [11].

Recently, multiple intradiol dioxygenases have been described in *A. niger*. Currently, four dioxygenases have been studied:, the catechol 1,2-dioxygenase (CrcA), hydroxyquinol 1,2-dioxygenase (HqdA), protocatechuic 3,4-dioxygenase (PrcA) and NRRL3_05330, a putative hydroxyquinol 1,2-dioxygenase [22,28,29,30]. Also, a methoxyhydroquinone 1,2-dioxygenase (MhdA) was identified that shares similarities with homogentisate 1,2-dioxygenases [81]. The role of NRRL3_05330 remains unknown since deletion did not result in any phenotypes [22].

Protocatechuic acid is one of the main intermediates in the degradation of hydroxy phenolic compounds. It has been shown for several Aspergilli that the aromatic ring of protocatechuic acid is cleaved into 3-carboxy-*cis,cis*-muconic acid and further converted through the β-ketoadipate pathway into acetyl-CoA and succinyl-CoA [76,100]. Several genes in this pathway were identified in *A. nidulans* and deletion of these genes resulted in reduced growth on protocatechuic acid [76]. More recently, the β-ketoadipate pathway has been further studied in *A. niger*, resulting in the identification of four enzymes (3-carboxy-*cis,cis*-muconate cyclase (CmcA; NRRL3_02586), 3-carboxymuconolactone hydrolase/decarboxylase (ChdA; NRRL3_01409), β-ketoadipate:succinyl-CoA transferase (KstA; NRRL3_01886) and β-ketoadipyl-CoA thiolase (KctA; NRRL3_01526)) [100]. In addition, an essential protein (NRRL3_00837) was found to be involved in the β-ketoadipate pathway, but its function remains unknown [100].

In *A. nidulans* and *A. niger*, catechol is cleaved by CrcA into *cis,cis*-muconic acid, with subsequential conversion toward 3-oxoadipate. In *A. nidulans*, several genes involved in the degradation of catechol have been identified via whole-genome transcriptomics and proteomics [76]. Deletion of AN3895 (muconate isomerase), AN4061 (muconolactone isomerase), and AN4531 (3-oxoadipate enol-lactone hydrolase) results in the formation of *cis,cis*-muconic acid, muconolactone and 3-oxoadipate enol-lactone, respectively, when grown on salicylic acid. Homologs of these genes were also identified in *A. niger* via whole-genome transcriptomics and were strongly induced by salicylic acid [22].

## 10. Regulation of Aromatic Metabolic Pathways

Transcriptional regulators are important for the regulation of metabolic pathways in fungi. Many transcription factors involved in the regulation of sugar metabolic pathways have been described [102]. However, only a few transcriptional regulators involved in the regulation of metabolic pathways for aromatic compounds have been identified. Recently, a regulator/repressor (TanR/TanX) complex involved in the regulation of gallic acid metabolic genes has been identified [92]. Whole-genome transcriptomic data concerning *A. niger* Δ*tanX* grown on fructose revealed that *phyA* is 2300 times more expressed than the parental strain. As mentioned previously, the cinnamic/sorbic acid metabolic pathway genes *cdcA* and *padA* are regulated by SdrA. More recently, it was shown that the weak acid regulator A and B (WarA and WarB) also play a role in the degradation of cinnamic acid and sorbic acid; however, the exact role of WarA and WarB needs to be further studied [103].

Fatty acid regulator A (FarA) has a role in the degradation of fatty acids [104,105] but also hydroxycinnamic acids in *A. niger* [106]. Deletion of *farA* results in reduced growth on *p*-coumaric acid and halted growth on ferulic acid and caffeic acid and the fatty acids valeric acid and oleic acid. A new far-like protein (FarD) was identified and deletion of it results in the cessation of growth on caffeic acid and ferulic acid and reduced growth on *p*-coumaric acid and *p*-hydroxybenzoic acid. Deletion of *farA* in *A. nidulans* results in reduced expression of *foxA* [104], which is also important for the degradation of hydroxycinnamic acids [36]. A putative FarA binding site has been observed in the promoter of *hscA* [106]. The exact role of FarA in the degradation of aromatic compounds needs to be studied further, but it is highly likely that it regulates genes involved in the CoA-dependent beta-oxidative metabolic pathway.

Recently, it was observed that the regulator FarB has a role in the degradation of benzoic acid, vanillic acid and the short fatty acid valeric acid, but not in the degradation of *p*-coumaric acid, caffeic acid, protocatechuic acid or *p*-hydroxybenzoic acid [106]. Currently, it remains unknown which genes are regulated by FarB. However, it is possible that FarB regulates *bphA* since reduced growth was observed on benzoic acid but not on *p*-hydroxybenzoic acid and protocatechuic acid, which correlates to the *bphA* deletion phenotype (Table 1).

## 11. Future Perspectives: Strategies for Studying and Identifying Aromatic Metabolic Pathways in Filamentous Fungi

The aromatic metabolic pathways of filamentous fungi belonging to the *Aspergillus* family have been better-studied compared to other fungal species. However, many fungal aromatic metabolic pathways still have gaps or remain unstudied. In addition, many of the enzymes involved in these pathways remain to be identified.

To study the aromatic metabolic pathways in fungi, deletion of the ring-cleaving dioxygenases is a straightforward approach, as these enzymes play critical roles in the metabolic pathway and deleting them causes severe phenotypes on aromatic compounds (Table 2) [11,76,81,92]. In *A. niger*, deletion of *prcA* led to clear phenotypes on many aromatic compounds related to the H-unit, while deletion of *hqdA* alone did not result in any phenotypes on the tested aromatic compounds [29]. However, the combined deletion of *prcA* and *hqdA* resulted in more severe phenotypes and even revealed the existence of an alternative pathway in *A. niger*. Deletion of *mhdA*, revealed that aromatic compounds related to the G-unit are degraded to methoxyhydroquinone [81]. This pathway is important in relation to creating aromatic compounds such as vanillin and vanillic acid. Deletion of *crcA* revealed that salicylic acid and 2,3-dihydroxybenzoic acid are degraded to catechol and that there is no alternative enzyme or pathway present in *A. niger* [22]. It appears that in *A. niger*, *ortho*-hydroxylated benzoic acids are converted into catechol, while *meta*- and *para*-hydroxylated benzoic acids are converted into protocatechuic acid and hydroxyquinol. *Meta*-methoxylated compounds such as ferulic acid, vanillic acid and veratric acid are converted into methoxyhydroquinone.

For the identification of genes encoding aromatic-compound-converting enzymes, whole-genome transcriptomics is a strong tool that can aid in the identification of genes after exposure to aromatic compounds. Currently, most genes involved in the aromatic metabolism of *A. niger* and *A. nidulans* have been identified using this approach [21,29,36,76,81,92]. In *A. niger*, a short exposure of 2 h to an aromatic compound can result in a strong induction of aromatic compound metabolic genes. For example, the salicylic acid metabolic genes *shyA*, *dhbA* and *crcA* are strongly upregulated by salicylic acid and by not *p*-hydroxyphenyl or guaiacylic aromatic compounds such as benzoic acid or *p*-coumaric acid, ferulic acid or vanillic acid. Similar observations were made for *vhyA* and *mhdA*, which are strongly induced by guaiacylic aromatic compounds, while the CoA-dependent β-oxidative genes *hcsA*, *foxA*, *katA*, *theA*, *theB*, *theC* and *theD* are strongly induced by hydroxycinnamic acids. This demonstrated that these genes are not only rapidly upregulated but also induced by specific aromatic compounds.

## 12. Future Perspectives: Production and Accumulation of Aromatic Compounds Using Fungi

As previously discussed, the aromatic metabolic pathways and the genes encoding aromatic-converting enzymes are slowly being identified (Table 1). Therefore, there are not many examples demonstrating the use of fungi in creating aromatic compounds. As vanillin cannot be produced by *A. niger* (Figure 1 and Figure 3), a two-step bioconversion with *A. niger* and the basidiomycete *Pycnoporus cinnabarinus* was used to demonstrate that vanillin can be produced from an agricultural waste stream, sugar beet pulp [107]. In this process, ferulic acid was released from the sugar beet pulp using heat, pressing, decantation and enzymatic treatments, resulting in 1 g of ferulic acid from 1 kg of dry sugar beet pulp. In the first step, precultured *A. niger* was grown with 900 mg of extracted ferulic acid for 4 days. The highest yield was observed at day 6, resulting in the formation of approx. 350 mg/L (50% molar yield) of vanillic acid and 85 mg/L (13.5% molar yield) of methoxyhydroquinone. The medium containing the produced vanillic acid was than filtered and used for the second step. In this step, *P. cynnabarius* was added to the medium to convert vanillic acid (150 mg/L) into vanillin (approx. 100 mg/L, 80% molar yield). It was observed that in the first step in this process, vanillic acid is converted into methoxyhydroquinone and degraded as a carbon source. This process could now be optimized in *A. niger*, with the deletion of *mhdA* resulting in the blockage of this conversion and therefor a higher vanillic acid yield [81]. In addition, the usage of *P. cinnabarinus* could also be removed by introducing a carboxylic acid reductase that converts vanillic acid into vanillin in *A. niger* and the deletion of *vdhA* to prevent the conversion of vanillin into vanillic acid. The highest reported vanillin production to date, using microorganisms, has been achieved using a genetically optimized strain of *Amycolatopsis* sp. (ATCC 39116) capable of converting 49.5 g of ferulic acid into 36.8 g of vanillin, corresponding to an approximate molar yield of 95% [108]. Currently, no genetically optimized fungi capable of vanillin production have been reported.

Another opportunity lies in the ability to accumulate aromatic compounds. Deletion of *prcA* and *phyA* showed that many H-unit-related aromatic compounds can be converted and accumulated into protocatechuic acid [28]. In addition, deletion of *vhyA* showed that many G-unit-related aromatic compounds can be converted into vanillic acid [81]. This means that complex mixtures of aromatic compounds can be converted into protocatechuic acid or vanillic acid. It has been demonstrated that a mixture of benzoic acid, benzaldehyde, caffeic acid, *p*-coumaric acid and *p*-hydroxybenzoic acid was efficiently converted into protocatechuic acid [28]. This ability is useful for processing complex mixtures of aromatic compounds, like those from depolymerized lignin, by funneling them into a single aromatic product.

## 13. Conclusions

In this review, recent advances in the understanding of aromatic metabolism in *A. niger* are highlighted. It is important to note that *Aspergillus* species, in general, possess a remarkable diversity of aromatic metabolic pathways, reflecting their adaptability to different environmental conditions and substrates. Given the already substantial versatility observed in *Aspergillus* species, it is anticipated that even greater metabolic diversity will be uncovered when studying basidiomycetes, which are known for their complex and highly specialized systems for degrading aromatic compounds such as lignin or other aromatic polymers. Understanding the metabolic pathways also creates new opportunities for the synthesis of aromatic compounds using fungi. Currently, the first steps toward creating an aromatic-compound-producing fungal cell factory are being taken, but it is clear that substantial knowledge gaps remain, particularly in terms of the identification of the genes and enzymes underlying many of these metabolic pathways.

## Figures and Tables

**Figure 1 microorganisms-13-01718-f001:**
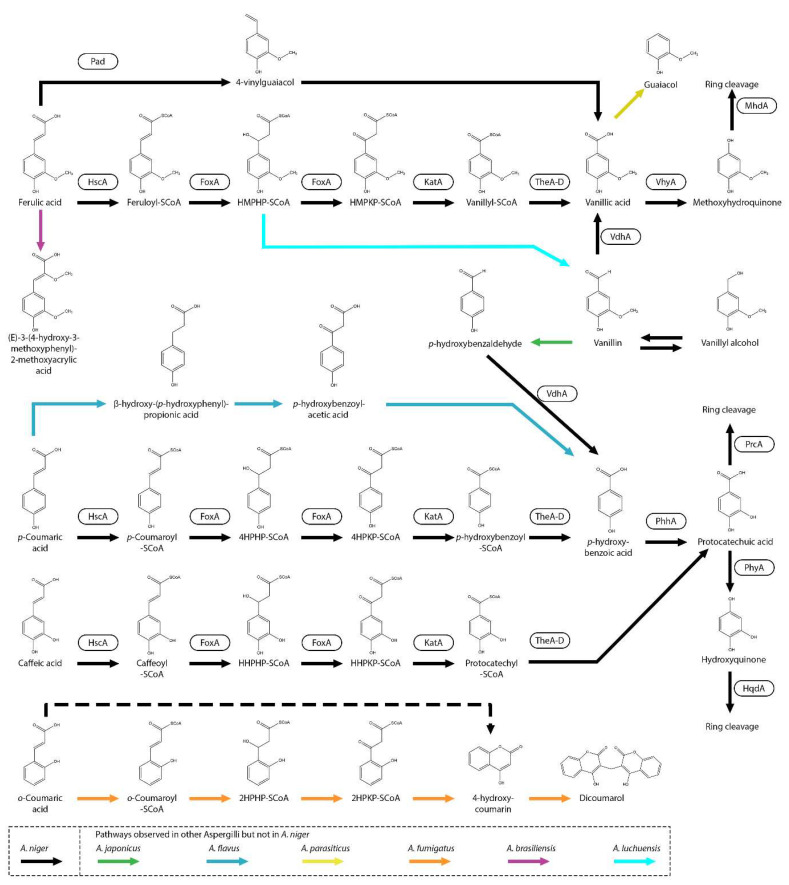
Metabolic pathway of the hydroxycinnamic acids, ferulic acid, *p*-coumaric acid and caffeic acid in *A. niger*. Pathways observed in *A. niger* are marked with black arrows. Unidentified pathways in *A. niger* are marked with dashed arrows. Boxes near an arrow represent identified enzymes catalyzing the reactions. Pathways that were not observed in *A. niger* but in other Aspergilli are marked with blue (*A. flavus*), green (*A. japonicus*) yellow (*A. parasiticus*), orange (*A. fumigatus*), cyan (*A. luchuensis*) and purple (*A. brasilliensis*) arrows. HMPKP-SCoA, 4-hydroxy-3-methoxyphenyl-β-hydroxypropionyl-CoA; HMPKP-SCoA, 4-hydroxy-3-methoxyphenyl-β-ketopropionic acid-CoA; 4HPKP-SCoA, 4-hydroxyphenyl-β-hydroxypropionyl-CoA; 4HPKP-SCoA, 4-hydroxyphenyl-β-ketopropionic acid-CoA; HHPKP-SCoA, 3,4-dihydroxyphenyl-β-hydroxypropionyl-CoA; HHPKP-SCoA, 3,4-dihydroxyphenyl-β-ketopropionic acid-CoA.

**Figure 2 microorganisms-13-01718-f002:**
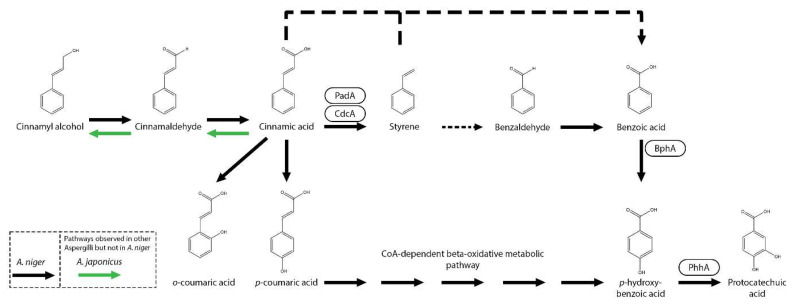
Cinnamic acid metabolic pathway in *A. niger*. Boxes near an arrow represent the enzymes catalyzing the reactions. Suggested pathways are marked with dashed arrows. Pathways that were not observed in *A. niger* but in other Aspergilli are marked with green (*A. japonicus*) arrows.

**Figure 3 microorganisms-13-01718-f003:**
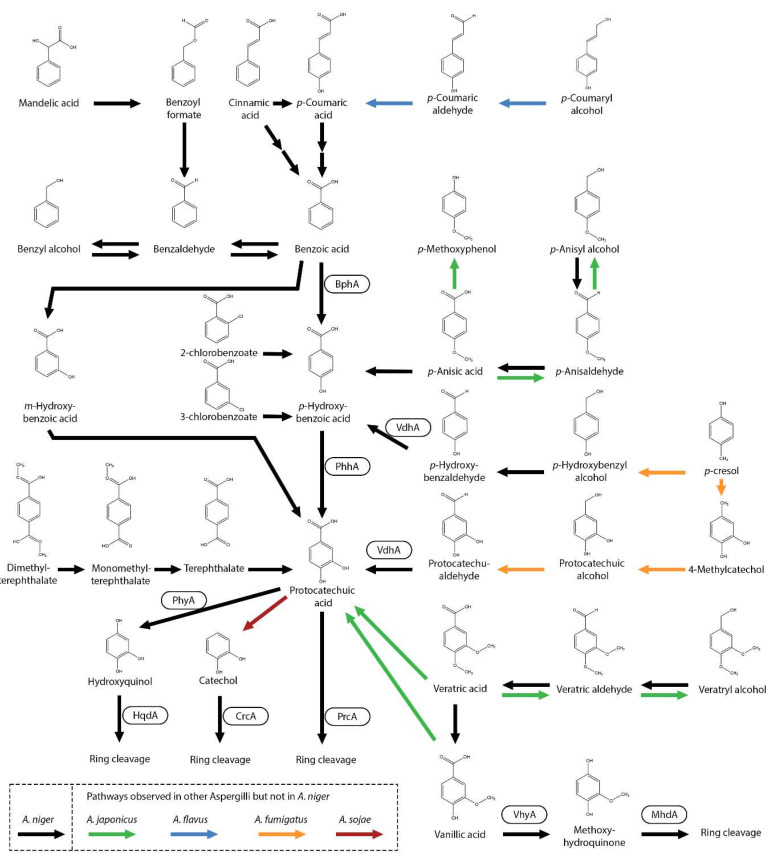
Benzoic acid and related compounds metabolic pathway in *A. niger*. Black arrows represent observed *A. niger* metabolic pathways and boxes near an arrow represent the enzymes catalyzing the reactions. Pathways that were not observed in *A. niger* but in other Aspergilli are marked with green (*A. japonicus*), blue (*A. flavus*), orange (*A. fumigatus*) or red (*A. sojae*) arrows.

**Figure 4 microorganisms-13-01718-f004:**
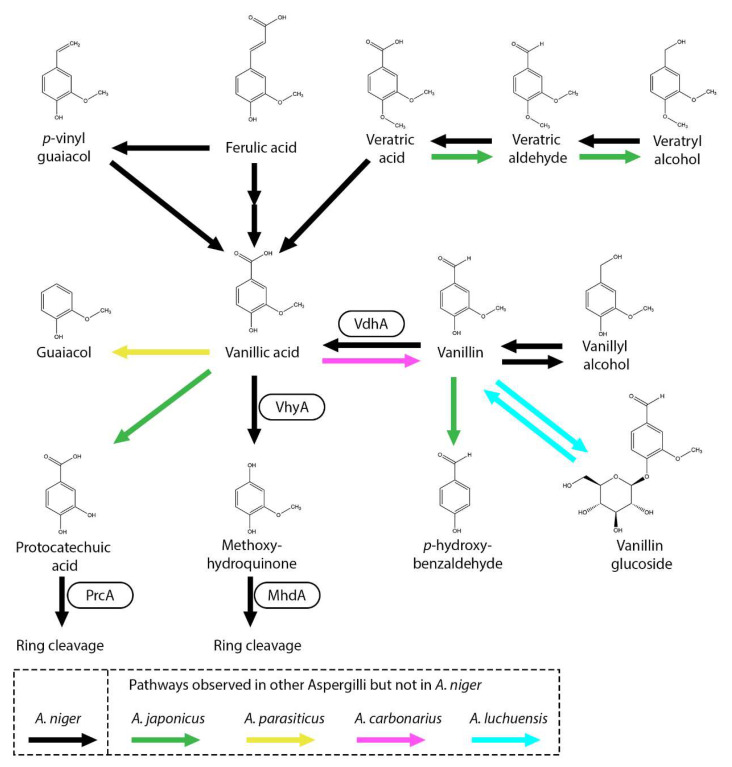
Vanillic acid and related compounds metabolic pathway in *A. niger*. Black arrows represent pathways observed in *A. niger*. Boxes near an arrow represent the enzymes catalyzing the reactions. Pathways that were not observed in *A. niger* but in other Aspergilli are marked with green (*A. japonicus*), pink (*A. carbonarius*), Cyan (*A. luchuensis*) and yellow (*A. parasiticus*) arrows.

**Figure 5 microorganisms-13-01718-f005:**
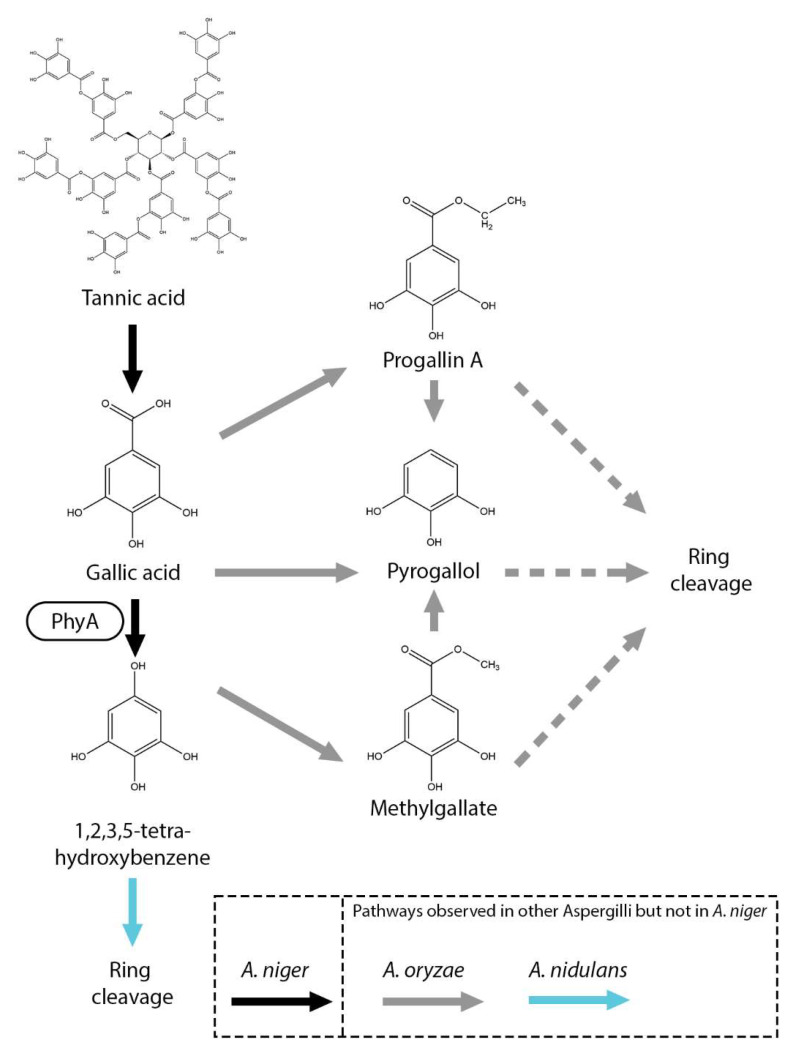
Gallic acid and related compounds metabolic pathway in *A. niger*. Black arrows represent pathways observed in *A. niger*. Boxes near an arrow represent the enzymes catalyzing the reactions. Suggested pathways are marked with dashed arrows. Pathways that were not observed in *A. niger* but in other Aspergilli are marked with gray (*A. oryzae*) and light blue (*A. nidulans*) arrows.

**Figure 6 microorganisms-13-01718-f006:**
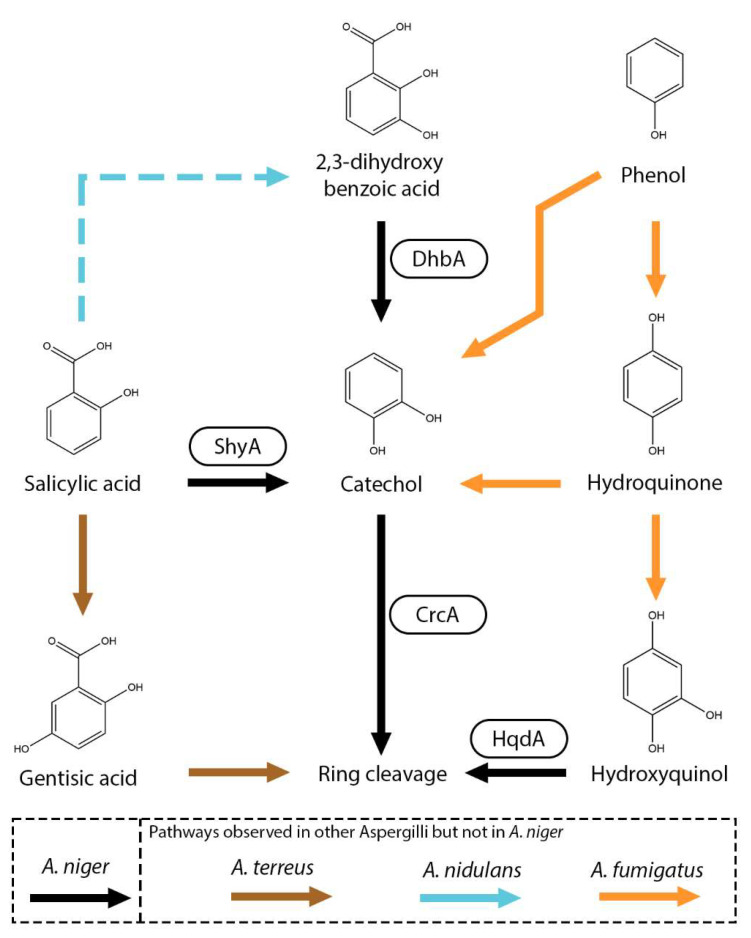
Salicylic acid metabolic pathway in *A. niger*. Boxes near an arrow represent the enzymes catalyzing the reactions. Suggested pathways are marked with dashed arrows. Pathways that were not observed in *A. niger* but in other Aspergilli are marked with brown (*A. terreus*), orange (*A. fumigatus*) or light blue (*A. nidulans*) arrows.

**Table 1 microorganisms-13-01718-t001:** Aromatic-converting enzymes identified in *Aspergillus* species.

Enzyme	*Aspergillus* Species	Year Identified	Evidence ^1^	Phenotype Observed on Aromatic Compounds When the Gene Is Deleted	Observed Enzymatic Activity of the Recombinant Protein Toward Aromatic Compounds	References
Benzoate-*p*-hydroxylase (BphA)	*A. niger*, *A. nidulans*	1990	IDA, IEP, IMP	Benzoic acid, cinnamic acid, benzaldehyde, benzyl alcohol	Benzoic acid, 2-fluorobenzoic acid, 2-chlorobenzoic acid, salicylic acid, 2-methylbenzoic acid, 3-fluorobenzoic acid, 3-chlorobenzoic acid, 3-methylbenzoic acid, 3-methoxybenzoic acid	[18,19,20,21]
2,3-dihydroxybenzoate decarboxylase (DhbA)	*A. niger*, *A. nidulans*	1995	IDA, IEP, IMP	2,3-Dihydroxybenzoic acid	2,3-Dihydroxybenzoic acid	[21,22,23,24]
Cinnamic acid decarboxylase (CdcA) Ferulic acid decarboxylase (FdcA)	*A. niger*	2010	IDA, IEP, IMP	Cinnamic acid	Cinnamic acid, caffeic acid, ferulic acid, *p*-coumaric acid	[25,26,27]
*p*-hydroxybenzoate-*m*-hydroxylase (PhhA)	*A. niger*, *A. nidulans*	2015	IDA, IEP, IMP	Benzoic acid, benzaldehyde, benzyl alcohol, cinnamic acid*,* *p*-hydroxybenzoic acid, *p*-coumaric acid, *p*-anisic acid, *p*-anisyl alcohol,	*p*-Hydroxybenzoic acid	[21,28,29]
Protocatechuate 3,4-dioxygenase (PrcA)	*A. niger*, *A. nidulans*	2015	EXP, IDA, IEP, IMP	Benzoic acid, benzaldehyde, benzyl alcohol, cinnamic acid, *m*-hydroxybenzoic acid, *p*-anisic acid, *p*-anisyl alcohol, *p*-coumaric acid, *p*-hydroxybenzoic acid, protocatechuic acid, protocatechuic aldehyde	Protocatechuic acid	[21,28,29,30]
Phenolic acid decarboxylase (PadA)	*A. luchuensis*	2018	IDA, IEP	n.d.	Caffeic acid, ferulic acid, *p*-coumaric acid	[31,32,33]
Protocatechuate hydroxylase (PhyA)	*A. niger*, *A. nidulans*	2021	IDA, IEP, IMP	Protocatechuic acid, gallic acid	Protocatechuic acid	[28,29]
Vanilliate hydroxylase (VhyA)	*A. niger*	2021	EXP, IDA, IEP, IMP	Ferulic acid, vanillic acid, vanillin	Vanillic acid	[34]
Vanillin dehydrogenase (VdhA)	*A. niger*	2021	IDA, IEP, IMP	Vanillin, *p*-hydroxybenzaldehyde, protocatechuic aldehyde	Vanillin, benzaldehyde, *p*-hydroxybenzaldehyde, protocatechuic aldehyde, syringic aldehyde, cinnamyl aldehyde, *p*-anisyl aldehyde, veratryl aldehyde	[34,35]
Methoxyhydroquinone 1,2-dioxygenase (MhdA)	*A. niger*	2021	IDA, IEP, IMP	Coniferyl alcohol, ferulic acid, vanillic acid, vanillin, vanillyl alcohol, veratric acid	Methoxyhydroquinone	[34]
Salicylate hydroxylase (ShyA)	*A. niger*, *A. nidulans*	2021	EXP, IDA, IEP, IMP	Salicylic acid	4-Aminosalicylic acid, 2,3-dihydroxybenzoic acid, gentisic acid, salicylic acid	[21,22]
Catechol 1,2-dioxygenase (CrcA)	*A. niger*, *A. nidulans*	2021	EXP, IDA, IEP, IMP	Salicylic acid, 2,3-dihydroxybenzoic acid, catechol	Catechol, hydroxyquinol, 3-methylcatechol, 4-methylcatechol	[21,22,30]
Hydroxycinnamate-CoA synthase (HcsA)	*A. niger*	2021	IEP, IMP	Ferulic acid, *p*-coumaric acid, caffeic acid, *m*-coumaric acid, dihydroferulic acid, dihydrocaffeic acid, phloretic acid	n.d.	[36]
Fatty acid oxidase (FoxA)	*A. niger*	2021	IEP, IMP	Ferulic acid, *p*-coumaric acid, caffeic acid, *m*-coumaric acid, dihydroferulic acid, dihydrocaffeic acid, phloretic acid	n.d.	[36]
3-ketoacyl-CoA thiolase (KatA)	*A. niger*	2021	IEP, IMP	Ferulic acid, *p*-coumaric acid, caffeic acid, *m*-coumaric acid, dihydroferulic acid, dihydrocaffeic acid, phloretic acid, 4-methoxycinnamic acid	n.d.	[36]
Thioesterases (TheA, TheB, TheC, TheD)	*A. niger*	2021	IEP, IMP	Ferulic acid, *p*-coumaric acid, caffeic acid, *m*-coumaric acid, dihydroferulic acid, dihydrocaffeic acid, phloretic acid	n.d.	[36]

^1^ Evidence is based on experimental assays. IDA, inferred from a direct assay using the enzyme in a biochemical assays; EXP, inferred from an accumulation experiment; IEP, inferred from the expression patterns; IMP, inferred from the mutant phenotype. n.d. not determined.

**Table 2 microorganisms-13-01718-t002:** Phenotypes of *A. niger* dioxygenase deletion transformants grown on monomeric aromatic compounds as the sole carbon source.

	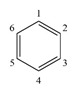							
Aromatic Compound	1	2	3	4	5	6	Δ*prcA* [65]	Δ*hqdA*[65]	Δ*prcA*/*hqdA*[65]	Δ*crcA*[34]	Δ*mhdA*[81]	Δ5330[34]	Δ17[92]
H-unit-related compounds
Cinnamic acid	CH=CHC=OOH	H	H	H	H	H	++	−	++	−	−	−	n.d.
p-coumaric acid	CH=CHC=OOH	H	H	OH	H	H	+	−	++	−	−	−	++
Caffeic acid	CH=CHC=OOH	H	OH	OH	H	H	+	−	++	−	−	−	++
Benzaldehyde	CH=O	H	H	H	H	H	++	n.d.	n.d.	n.d.	n.d.	n.d.	n.d.
Benzoic acid	C=OOH	H	H	H	H	H	++	−	++	n.d.	−	n.d.	n.d.
Benzyl alcohol	CH_2_OH	H	H	H	H	H	++	n.d.	n.d.	n.d.	n.d.	n.d.	n.d.
p-hydroxybenzoic acid	C=OOH	H	H	OH	H	H	++	−	++	n.d.	−	n.d.	++
p-hydroxybenzaldehyde	CH=O	H	H	OH	H	H	++	−	++	n.d.	n.d.	n.d.	n.d.
m-hydroxybenzoic acid	C=OOH	H	OH	H	H	H	++	n.d.	n.d.	n.d.	−	n.d.	n.d.
Protocatechuic acid	C=OOH	H	OH	OH	H	H	+	−	++	−	−	−	++
Protocatechuic aldehyde	CH=O	H	OH	OH	H	H	+	−	++	n.d.	−	n.d.	n.d.
p-anisic acid	C=OOH	H	H	OCH_3_	H	H	++	n.d.	n.d.	n.d.	n.d.	n.d.	n.d.
p-anisyl alcohol	CH_2_OH	H	H	OCH_3_	H	H	++	n.d.	n.d.	n.d.	n.d.	n.d.	n.d.
G-unit-related compounds
Coniferyl alcohol	CH=CHCH_2_=OH	H	OCH_3_	OH	H	H	n.d.	n.d.	n.d.	n.d.	++	n.d.	n.d.
Ferulic acid	CH=CHC=OOH	H	OCH_3_	OH	H	H	−	−	−	−	++	−	−
Vanillic acid	C=OOH	H	OCH_3_	OH	H	H	−	−	−	n.d.	++	n.d.	−
Vanillin	CH=O	H	OCH_3_	OH	H	H	n.d.	n.d.	n.d.	n.d.	++	n.d.	n.d.
Vanillyl alcohol	CH_2_OH	H	OCH_3_	OH	H	H	n.d.	n.d.	n.d.	n.d.	++	n.d.	n.d.
Veratric acid	C=OOH	H	OCH_3_	OCH_3_	H	H	−	n.d.	n.d.	n.d.	+	n.d.	n.d.
S-unit-related compounds
Gallic acid	C=OOH	H	OH	OH	OH	H	−	−	−	n.d.	n.d.	n.d.	−
Others
Salicylic acid	C=OOH	OH	H	H	H	H	−	−	n.d.	++	n.d.	−	++
2,3-dihydroxybenzoic acid	C=OOH	OH	OH	H	H	H	−	−	n.d.	++	n.d.	−	n.d.
Catechol	OH	OH	H	H	H	H	−	−	−	++	n.d.	−	++
Gentisic acid	C=OOH	OH	H	H	OH	H	n.d.	n.d.	n.d.	n.d.	n.d.	n.d.	−

−, no phenotype; +, phenotype; ++, strong phenotype; n.d., not determined.

## Data Availability

Not applicable.

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
