# Peer review of "An Updated Perspective on the Aromatic Metabolic Pathways of Plant-Derived Homocyclic Aromatic Compounds in Aspergillus niger"

_microorganisms, 2025, doi:10.3390/microorganisms13081718_

Round 1
Reviewer 1 Report
Comments and Suggestions for Authors
Dear Author,
first of all many congrats toward the structure of MS, especially concerning metabolomic schema which are of the most important results in understanding of this difficult topic about understanding fungal aromatic metabolic pathways in Aspergilli species sensus Nigri section. Since this review is focused more on biochemistry, just few facts about the biology of this fungi is essential in the first part of the MS.
Moreover, I have one suggestion for you to make one review table in order to present it in more clear way in the first part of the MS. Just think about that possibillity, it is not essential...
All changes have been incorporated in the MS and notified in comments.
Sincerely yours
Reviewer

Author Response
Dear reviewer,
Thank you for your time reviewing my manuscript. All answers on your comments are attached in the file and written with a blue font. Changes in the manuscript are highlighted in yellow.

Reviewer 2 Report
Comments and Suggestions for Authors
The review article gives an overview and comparison of different aromatic metabolic pathways of the plant derived heterocyclic aromatic compounds in Aspergillus niger. Overall, the author has made a great effort and presented it well in terms of figures and identifying the gap. It seems the author has a sound knowledge about the topic. However, the biosynthesis and degradation of aromatic compounds are two different complex systems and the lignin depolymerization and its conversion to monomeric aromatic compounds itself as a challenging step. In lignin to aromatic conversion and then further aromatic catabolism, there are several up and down pathways. The author directly linked the plant derived lignin source to the aromatic compounds; however, the author didn’t discuss about lignin conversion to those compounds and the importance of fungi in lignin degradation and aromatic bio funneling, leaving a big gap there. In my opinion, it is better to understand the process of producing the aromatic compounds from lignin first. Besides, the conclusion was a bit confusing, whether the article focused on aromatic compounds degradation or its biosynthesis. Besides from the scientific point of view, the manuscript has issues in grammatic mistakes, scientific terminologies and nomenclature. While this review has some merits, I recommend a major revision to address the issues. Some specific comments that could improve the manuscript are as below.
Line 16: Can fungi produce bioplastics, could you name any bioplastic that is produced by fungi?
Line 21-26: Why do you think that understanding aromatic metabolism in fungi is essential, what are the advantages of aromatic bioconversion in fungi compared to bacteria?
Line 59: “in order the fill up the gaps” change as “in order to”
Line 63: TLC and GLC need to be defined on the first appearance.
Line 82: “on complex plant biomass” should be revised as “of complex plant biomass”.
Line 85, 93: Abbreviated genus name should be used after the first appearance for presenting accurate scientific nomenclature. The full name was described before. See line: 22
Line 99: observed other fungi. In is missing (Preposition mistake).
Line 117: The author needs to write full names of the para and meta coumaric acid. Other similar abbreviations given on first appearance may need to be revised throughout the manuscript.
Line 226: but is in A. niger converted to vanillic acid?? Grammatical mistake
Line 265, 273: Nomenclature issue
Line 271. Citation style issue
Line 109, 136, 298, 373, 435, 446, 448: The word “abolished” was written many times, why not to say resulted in growth suppression or reduced growth. Does the author think it is a more suitable word?
Line 433-434, 435-437: These statements have some confusions and irregular repetition of words were found. I suggest revising it to avoid confusion.
Line 445-446: Can the author clarify the difference b/w abolished growth and reduced growth?
Line 457-458: “no phenotypes are not observed”. wrong statement
Line 463-464: Wrong statement.
Table 1: Captions need to be revised
Table 1: Some citation font size is too small. Resize it according to the font size of the manuscript.
References: Some of the references have no page numbers such as, 2, 5, 24, 51, 54, 81, 86, 96, 97, 98.
Author Response

(The authors gave the same response as above.)

Reviewer 3 Report
Comments and Suggestions for Authors
The manuscript An Updated Perspective on the Aromatic Metabolic Pathways of Plant-Derived Heterocyclic Aromatic Compounds in Aspergillus niger by R.J.M. Lubbers examines the processing of plant metabolites by enzymes of the fungus Aspergillus niger.
The manuscript is formatted according to the rules and contains all the necessary sections.
The materials are well formatted and cover a wide range of processes of transformation of plant molecules, mainly lignin.
I believe that a clearer description of the fungal targets and a description of the conditions under which the enzymes exhibit optimal activity are essential.
It is also necessary to list specific substrates, their origin and properties suitable for the cultivation of Aspergillus niger. The main thing is the source of the substrate, since it is not very clear in the text whether substances from outside or substances synthesized by the fungus itself are metabolized. Another issue is the lack of information on directed mutagenesis, genetic engineering, and genotype diversity in lignin conversion.
Author Response

(The authors gave the same response as above.)

Reviewer 4 Report
Comments and Suggestions for Authors
The review manuscript entitled “An Updated Perspective on the Aromatic Metabolic Pathways of Plant-Derived Heterocyclic Aromatic Compounds in Aspergillus niger” by R.J.M. Lubbers is a high-quality, well-written, and comprehensive review on the subject. I therefore recommend it for publication after minor revisions.
Minor revisions
Figure 3 p-coumaric acid is metabolized to benzoic acid or to p-hidroxybenzoic-acid? Is there evidence that benzoyl formate is formed directly from mandelic acid and not to an unspecific or detoxification product?
Figure 5 Shouldn’t the arrow from pyrogallol to ring cleavage be dashed (suggested but not confirmed)?
Line 59 please revise
Line 133 please revise
Line 271 revise reference /missing in references
Line 362 please revise
Author Response

(The authors gave the same response as above.)

Reviewer 5 Report
Comments and Suggestions for Authors
Review for the manuscript Microorganisms-3682268. An Updated Perspective on the Aromatic Metabolic Pathways of Plant-Derived Heterocyclic Aromatic Compounds in Aspergillus niger by R.J.M. Lubbers
Several remarks on the manuscript were appeared and are listed below.
Lines 13-19.
Lignin, a complex polymer in plant cell walls, is the largest renewable source of aromatic compounds, though its degradation remains challenging. … Microorganisms, including fungi, offer a sustainable alternative in breaking down lignin into valuable compounds. Fungi possess unique enzymes capable of converting aromatic compounds derived from lignin into biofuels, bioplastics, and chemical building blocks. The manuscript does not analyze breaking down lignin into valuable compounds, biofuels, bioplastics, and chemical building blocks. There are no any data on the possible yields of the most useful products of the lignin degradation by A. niger. For example, let us look at Fig. 1: how much ferulic acid can be obtained, how much vanillin?
However, their aromatic metabolic pathways are less studied compared to bacterial systems. There are no attempts to compare metabolic pathways for A. niger and bacterial systems.
Recent advances in genomics, proteomics, and metabolic engineering are helping to reveal these pathways, offering new opportunities to optimize fungi for lignin bioconversion.
Reviewer does not see offering new opportunities to optimize fungi for lignin bioconversion.
I do not see the goal of the made review in the manuscript.
Gene engineering is developing to produce useful products in high yields, for example, vanillin from lignin or ferulic acid. I understand these goals as interesting and useful. But the manuscript looks like simple listing the known metabolic pathway of hydroxycinnamic acids, ferulic acid and other monomers by Aspergilius.
Lines 49-53.
…in lignin [3]. Hydroxycinnamic acids such as ferulic acid (3-methoxy-4-hydroxycinnamic acid) and p-coumaric acid (p-hydroxycinnamic acid) can be linked to complex plant cell wall structures like hemi-cellulose and pectin and have ester linkages with arabinose and galactose [11– 13]. Currently, much research is performed in the utilization of lignin through chemical, enzymatic and biotransformation approaches [4–7]. References numeration is disrupted.
Lines 61-63.
Before 2020, the best described aromatic metabolic pathways were from Aspergillus japonicus and were obtained by using UV spectra, TLC, GLC and dioxygenase activity [9]. As for review, it does not matter, how were the results [9] obtained. The main question is – what results were obtained in [9].
Lines 71-73.
In this review, the recent advances and insight in the monomeric aromatic metabolic pathways in fungi from the past years, with a main focus on Aspergillus niger, are highlighted and discussed.
This goal does not correspond to the Abstract content (see above, Microorganisms, including fungi, offer a sustainable alternative in breaking down lignin into valuable compounds). Neither breaking down lignin nor valuable compounds and their yields are described in the manuscript.
Lines 452-454.
The exact role of FarA in the degradation of aromatic compounds need to be studied further but it is highly likely that it regulates multiple genes involved in the CoA-dependent beta-oxidative metabolic pathway. it is regulated by multiple genes?
Lines 457-459.
It is possible that FarB is regulating bphA since no phenotypes are not observed on p-hydroxybenzoic acid and protocatechuic acid. Double negative? …any phenotypes are not observed…?
Lines 513-518, Conclusion.
Additionally, a deeper understanding of regulatory networks and enzyme mechanisms involved in aromatic biosynthesis could lead to innovative bioprocesses, reducing reliance on petrochemical-based production and promoting sustainability. As research continues to uncover novel and expand fungal aromatic metabolic pathways, the potential for industrial applications will continue to expand, reinforcing fungi as valuable cell factories for aromatic compounds. To attain these prospects, the data on the products yield on the metabolic pathways are necessary, but the review ignores this necessity completely.
Author Response

(The authors gave the same response as above.)

Round 2
Reviewer 2 Report
Comments and Suggestions for Authors
The authors are advised to double check the final version of the manuscript in order to avoid any miss place citation and grammatical issues.
Author Response
The authors are advised to double check the final version of the manuscript in order to avoid any miss place citation and grammatical issues.
The manuscript was double checked for grammatical errors.
Reviewer 5 Report
Comments and Suggestions for Authors
Reviewer 5:
Review for the manuscript Microorganisms-3682268. An Updated Perspective on the Aromatic Metabolic Pathways of Plant-Derived Heterocyclic Aromatic Compounds in Aspergillus niger by R.J.M. Lubbers
Several remarks on the manuscript were appeared and are listed below.
Line 276. High protocatechuic acid accumulation rates, up to 90-99%, were obtained from most of the tested aromatic compounds. A rate is not measured in %, and % assumes comparison.
Lines 536-543. In this process, ferulic acid was released from sugar beet pulp using heat, pressing, decantation and enzymatic treatments, resulting 1 g ferulic acid from 1 kg dry sugar beet pulp. In the first step, precultured A. niger was grown with 900 mg extracted ferulic acid for 4 days. The highest yield was observed at day 6, resulting in the formation of approx. 350 mg/L (50% molar yield) vanillic acid and 85 mg/L (13.5% molar yield) methoxyhydroquinone. The medium containing the produced vanillic acid was than filtered and used for the second step. In this step, P. cynnabarius was added to the medium to convert vanillic acid (150 mg/L) into vanillin (approx. 100 mg/L, 80% molar yield). Please, compare and cite: Fleige C, Meyer F, Steinbüchel A. 2016. Metabolic engineering of the actinomycete Amycolatopsis sp. strain ATCC 39116 towards enhanced production of natural vanillin. Appl Environ Microbiol 82:3410–3419. doi:10.1128/AEM.00802-16. 20 g/l of vanillin from ferulic acid are synthesized.
Author Response
Line 276. High protocatechuic acid accumulation rates, up to 90-99%, were obtained from most of the tested aromatic compounds. A rate is not measured in %, and % assumes comparison.
Adjusted to: High protocatechuic acid accumulation rates, up to 90-99% molar yield, were obtained from most of the tested aromatic compounds.
Lines 536-543. In this process, ferulic acid was released from sugar beet pulp using heat, pressing, decantation and enzymatic treatments, resulting 1 g ferulic acid from 1 kg dry sugar beet pulp. In the first step, precultured A. niger was grown with 900 mg extracted ferulic acid for 4 days. The highest yield was observed at day 6, resulting in the formation of approx. 350 mg/L (50% molar yield) vanillic acid and 85 mg/L (13.5% molar yield) methoxyhydroquinone. The medium containing the produced vanillic acid was than filtered and used for the second step. In this step, P. cynnabarius was added to the medium to convert vanillic acid (150 mg/L) into vanillin (approx. 100 mg/L, 80% molar yield). Please, compare and cite: Fleige C, Meyer F, Steinbüchel A. 2016. Metabolic engineering of the actinomycete Amycolatopsis sp. strain ATCC 39116 towards enhanced production of natural vanillin. Appl Environ Microbiol 82:3410–3419. doi:10.1128/AEM.00802-16. 20 g/l of vanillin from ferulic acid are synthesized.
Added and cited the paper. ''The highest reported vanillin production to date, using microorganisms, has been achieved using a genetically optimized strain of Amycolatopsis sp. (ATCC 39116), capable of converting 49.5 g of ferulic acid into 36.8 g of vanillin, corresponding to an approximate molar yield of 95% [113]. Currently, no genetically optimized fungi capable of vanillin production has been reported.''